# Reducing nonradiative recombination for highly efficient inverted perovskite solar cells via a synergistic bimolecular interface

Shaobing Xiong[1,2,11], Fuyu Tian[3,11], Feng Wang ●[4,11], Aiping Cao[1], Zeng Chen[5], Sheng Jiang[1], Di Li[1], Bin Xu[1], Hongbo Wu[6], Yefan Zhang[7], Hongwei Qiao[1], Zaifei Ma ●[6], Jianxin Tang ●[7], Haiming Zhu ●[5], Yefeng Yao ●[1], Xianjie Liu[8], Lijun Zhang ●[3] ✉, Zhenrong Sun[9], Mats Fahlman ●[8], Junhao Chu[2], Feng Gao ●[4] ✉ & Qinye Bao ●[1,2,10] ✉

Reducing interface nonradiative recombination is important for realizing highly efficient perovskite solar cells. In this work, we develop a synergistic bimolecular interlayer (SBI) strategy via 4-methoxyphenylphosphonic acid (MPA) and 2-phenylethylammonium iodide (PEAI) to functionalize the perovskite interface. MPA induces an in-situ chemical reaction at the perovskite surface via forming strong P-O-Pb covalent bonds that diminish the surface defect density and upshift the surface Fermi level. PEAI further creates an additional negative surface dipole so that a more n-type perovskite surface is constructed, which enhances electron extraction at the top interface. With this cooperative surface treatment, we greatly minimize interface nonradiative recombination through both enhanced defect passivation and improved energetics. The resulting p-i-n device achieves a stabilized power conversion efficiency of 25.53% and one of the smallest nonradiative recombination induced $V_{oc}$ loss of only 59 mV reported to date. We also obtain a certified efficiency of 25.05%. This work sheds light on the synergistic interface engineering for further improvement of perovskite solar cells.

Inverted p-i-n perovskite solar cells (PSCs) possess remarkable advantages of low-temperature processibility, long-term stability, and compatibility in state-of-the-art tandem cells, making them promising in photovoltaic commercialization[1–3]. Especially, recent explorations on defect elimination, crystallization control and interface design via additive or surface reconstruction have improved their power conversion efficiency (PCE) to certified values over 25%[4–6]. Nevertheless, the nonradiative recombination at charge-extracting interfaces in the inverted p-i-n device can be further decreased to push the efficiency towards the Shockley-Queisser limit[7].

Surface defects due to interruption of the perovskite crystal lattice result in significant traps for photogenerated carriers that cause

[1]School of Physics and Electronic Science, East China Normal University, Shanghai 200241, China. [2]Shanghai Frontiers Science Research Base of Intelligent Optoelectronics and Perception, Institute of Optoelectronics, Fudan University, Shanghai 200433, China. [3]State Key Laboratory of Integrated Optoelectronics, Key Laboratory of Automobile Materials of MOE, International Center of Computational Method and Software, School of Materials Science and Engineering, Jilin University, Changchun 130012, China. [4]Department of Physics, Chemistry and Biology, Linköping University, Linköping 58183, Sweden. [5]Department of Chemistry, Chemistry of High-Performance and Novel Materials, Zhejiang University, Hangzhou 310027, China. [6]Center for Advanced Low-Dimension Materials, Donghua University, Shanghai 201620, China. [7]Institute of Functional Nano & Soft Materials (FUNSOM), Soochow University, Suzhou 215123, China. [8]Laboratory of Organic Electronics, Linköping University, Norrköping 60174, Sweden. [9]State Key Laboratory of Precision Spectroscopy, East China Normal University, Shanghai 200241, China. [10]Collaborative Innovation Center of Extreme Optics, Shanxi University, Taiyuan, Shanxi 030006, China. [11]These authors contributed equally: Shaobing Xiong, Fuyu Tian, Feng Wang. ✉e-mail: lijun_zhang@jlu.edu.cn; feng.gao@liu.se; qybao@clpm.ecnu.edu.cn

recombination loss, provide channels for ions migration and external environmental invasion and thus destroy device operational efficiency and stability[8,9]. Functional agents such as Lewis acid and fluoride are commonly applied to passivate perovskite surfaces by forming coordination bonds, ionic bonds or hydrogen bonds[10–12]. However, these resulting bonds are significantly weaker than that of primary covalent bonds used in e.g., silicon solar cells, limiting the passivation efficacy and durability[13,14]. Covalent bonds, which have great robustness and are expected greatly to affect surface chemical and physical properties, are not yet investigated in-depth for perovskite semiconductors.

Interface energetics largely control the charge extraction and recombination, affecting all the three photovoltaic parameters, i.e., open-circuit voltage ($V_{oc}$), fill factor (FF), and short-circuit current density ($J_{sc}$)[15–17]. Unmatched interface energetics will form a potential well and trap charge carriers at perovskite/charge transport layer (CTL) interfaces, reducing $V_{oc}$ and FF. The resulting charge accumulation is reported to reduce the activation energy for halide migration[18], so that a well-matched energy level alignment is required both for performance and stability enhancement.

These currently unsolved challenges emphasize the need to develop a molecularly functionalized interface providing the following critical functionalities: (i) covalent bonds for strong interface passivation and (ii) matched energetics that boost charge extraction into the CTL. However, design and synthesis of a single molecule that can provide both the desired covalent bonding and energy level alignment could be complicated. Co-depositing multiple surface-modifying molecules is also challenging, of which the competing interactions with the perovskite will not only limit the passivation efficiency, but also create an inhomogeneous energy landscape.

Here we develop a synergistic bimolecular interlayer (SBI) strategy to address the above two issues and achieve high-performance p-i-n PSCs. A 4-methoxyphenylphosphonic acid (MPA) modulator is employed to chemically solder the perovskite surface via strong P-O-M (metal) bonds, where M refers to $Pb^{2+}$ ions. The strong chemical interaction largely suppresses the interface recombination by reducing trap state density and upshifting Fermi level ($E_F$) of the perovskite surface. A second 2-phenylethylammonium iodide (PEAI) layer is then sequentially deposited to further modulate the interface energetics by forming a negative surface dipole, which constructs a more n-type perovskite surface and enhances electron extraction across the top interface. As a result, the SBI-based p-i-n PSC shows a stabilized efficiency of 25.53% and one of the smallest nonradiative recombination induced $V_{oc}$ loss of only 59 mV reported to date. In addition, the target device also features good stability, retaining 95% of its initial efficiency for aging over 1000 h at $55 \pm 5\,°C$.

## Results

### SBI strategy and surface reaction

The p-i-n device configuration is depicted in Fig. 1a, where MPA and PEAI are sequentially inserted between the perovskite $Cs_{0.05}(FA_{0.95}MA_{0.05})_{0.95}Pb(I_{0.95}Br_{0.05})_3$ and the electron transport layer (ETL) PCBM to form the SBI (chemical structures of the materials shown in Fig. 1b). The thickness of the SBI is estimated to be <6 nm (Fig. 1c and Supplementary Note 1). We find that the SBI does not adversely affect the perovskite surface morphology, crystallinity, or optical absorption properties (Supplementary Figs. 1–3). The band edge of the perovskite remains constant at 1.55 eV. The water contact angle measurement presents that the SBI-modified perovskite surface is more hydrophobic, beneficial for the long-term stability of PSCs (Supplementary Fig. 4). X-ray photoelectron spectroscopy (XPS) is conducted to demonstrate covalent bonding of perovskite surface with the deposition of MPA. The control perovskite film has an ideally sharp doublet Pb 4$f$ feature assigned to the chemical state of Pb-I/Br, a single N 1$s$ peak referred to $CH=NH_2^+$/$CH-NH_2$ of $FA^+$ and a minor signal O 1$s$ peak from absorbed oxygen (Fig. 1d–f). After MPA surface

modification, the Pb 4$f$ core level splits into two doublets at higher and lower binding energy (BE) compared to the original doublet feature. The intensities of the high BE peak increase with enhancing MPA concentrations (Supplementary Fig. 5a), which we tentatively attribute to the chemical interaction of uncoordinated $Pb^{2+}$ with deprotonated -PO(OH)O$^-$ in terms of P-O-Pb bonds[19]. The deprotonation of phosphonic acid is expected in its dissolution process[20]. As confirmed by the pH test (Supplementary Fig. 6), the ethanol solvent shows lower pH value when MPA is dissolved with more solvated protons, indicating the deprotonation of phosphonic acid in the MPA solution. The O 1$s$ signal represents a significant increase due to the $CH_3O$- (530.9 eV), P=O (532.6 eV), and P-O(H) (533.6 eV) from MPA. In comparison with the O 1$s$ chemical environment of pristine MPA (Supplementary Fig. 7), a new peak near 531.5 eV emerges and the dominating P-O(H) chemical state is decreased, consistent with the formation of P-O-Pb bonds and the deprotonation of -PO(OH)$_2$. Additionally, the Pb 4$f$, N 1$s$, O 1$s$, I 3$d$, P 2$p$ and C 1$s$ peaks all shift to lower BE due to the strong surface chemical reaction (Supplementary Fig. 5). We use Fourier-transform infrared (FTIR) spectra to further confirm the generation of P-O-Pb vibration signal near 1076 cm$^{-1}$, and the P-O(H) vibration peaks show obvious downward shifts (Supplementary Fig. 8) in MPA/$Cs_{0.05}(FA_{0.95}MA_{0.05})_{0.95}Pb(I_{0.95}Br_{0.05})_3$ films. As the P=O vibration remains unchanged, the MPA prefers to bind with the perovskite through strong covalent P-O-Pb bonds rather than the weak coordination (i.e., P=O → Pb) that has been widely adopted[21,22]. The achieved interfacial covalent P-O-M (metal) bonds are critical as they have great robustness and can significantly affect surface chemistry and physics of metals and metal oxides[23,24] including a more effective passivation of defects[13,25].

Furthermore, the introduction of the PEAI layer exhibits no obvious change to the chemical species of Pb 4$f$, O 1$s$, I 3$d$, and P 2$p$ XPS spectra (Fig. 1d–f and Supplementary Fig. 9). The shift to higher BE of the XPS peaks implies that the physically absorbed PEAI might create a downshifting dipole on the MPA-coated perovskite surface. In addition, it is also expected that PEAI provides additional defect passivation effectiveness to the perovskite surface due to the inconsecutive coverage of the MPA layer.

### Surface energetics

The impact of the SBI on the perovskite surface energetics is explored in detail by ultraviolet photoelectron spectroscopy (UPS). The work function (WF) is derived from the secondary electron cutoff, while the position of valence band maximum (VBM) relative to the $E_F$ is extracted from the onset of valence band in a logarithmic photoemission intensity scale (Fig. 2a and Supplementary Fig. 10). The surface WF of the MPA-modified perovskite film decreases from 4.54 to 4.32 eV, ascribed to charge redistribution by the strong surface reaction. The surface WF of the PEAI/MPA-modified perovskite film further decreases to 4.20 eV due to the formation of a surface dipole[17]. The corresponding VBM position shifts from 0.87 to 1.08 and 1.20 eV, respectively. As illustrated in Fig. 2b, the SBI largely tailors the perovskite surface energetics by moving the $E_F$ closer to the conduction band and constructs a more n-type perovskite surface, which is expected to introduce an extra built-in electric field and enhance electron extraction while blocking holes at the perovskite/ETL interface, thus also suppressing interface recombination[26,27]. In addition, as the resulting surface WF of the SBI-modified perovskite is smaller than the negative pinning level energy ($E_{ICT-} = 4.31$ eV) of the organic semiconductor PCBM used as ETL[28], the perovskite surface $E_F$ can pin to the negative polaron transport state of PCBM (Fig. 2c) and the perovskite thus creates a good contact with PCBM, further promoting electron transfer across the perovskite/ETL interface.

We utilize Kelvin probe force microscopy (KPFM) to examine the surface morphology and potential distribution of the perovskite film (Fig. 2d–g, Supplementary Fig. 11). The SBI-modified perovskite film

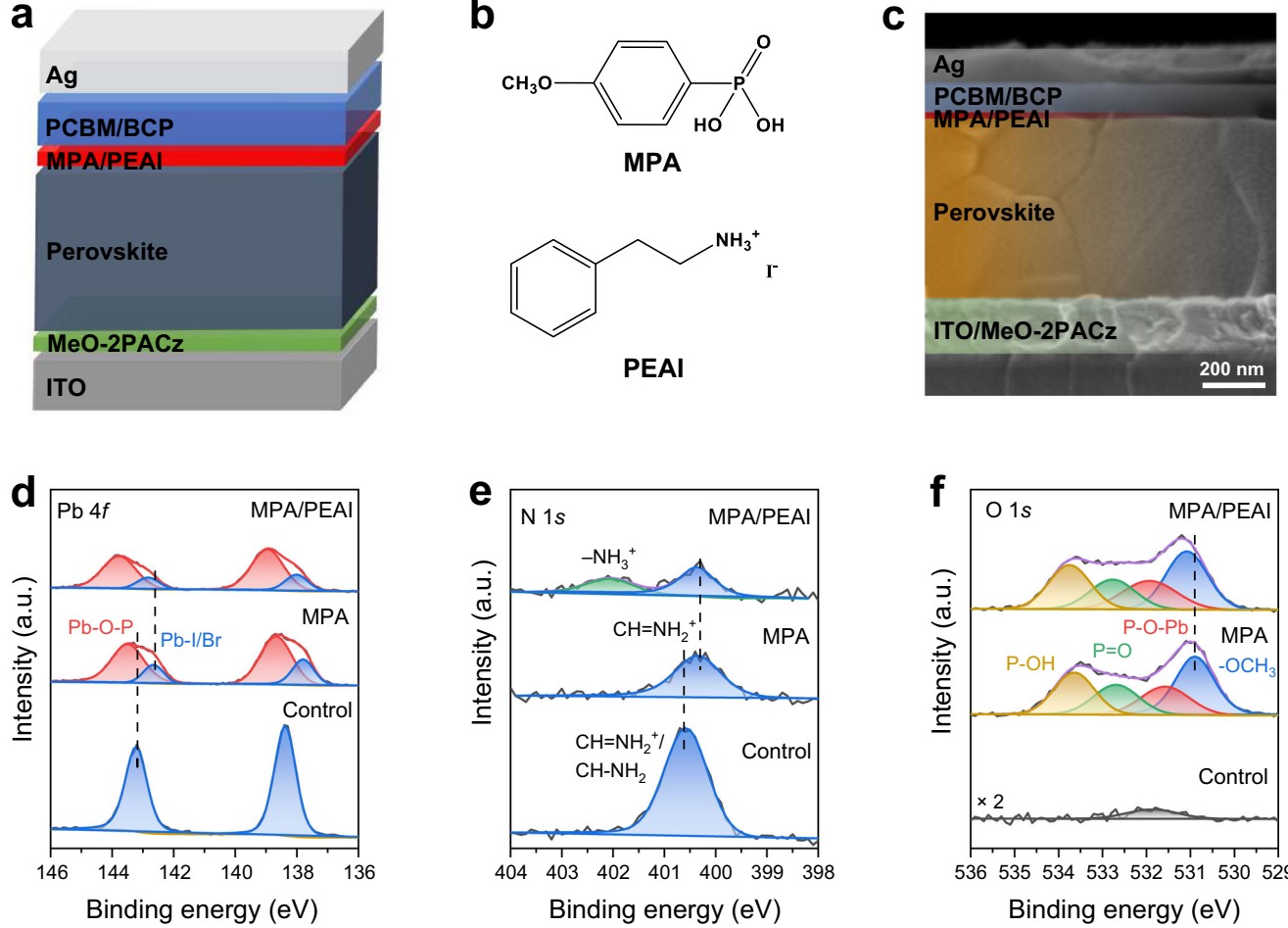

**Fig. 1 | SBI strategy and surface reaction. a** Schematic illustration of p-i-n PSC using MPA/PEAI as SBI. **b** Chemical structures of MPA and PEAI molecules. **c** A cross-sectional SEM image of PSC. **d–f** XPS core level characterizations of Pb, N, and O elements on the perovskite films with and without SBI.

exhibits a diminished contact potential relative to the control, in accordance with UPS results. Moreover, the SBI-modified perovskite film displays a smaller surface potential distribution difference and lower surface roughness. A smoother perovskite surface with a more uniform surface potential distribution is beneficial for forming an efficient contact with the adjacent ETL that prevents nonradiative recombination[16,29].

## Defect passivation and enhanced extraction

We conduct steady-state photoluminescence (PL) and time-resolved PL (TRPL) measurements to probe the effect of SBI on charge carrier dynamics. With the SBI, the PL intensity increases (Fig. 3a) and the corresponding TRPL average carrier lifetime increases by more than one time from 21.50 to 56.45 ns (Fig. 3b, c and Supplementary Table. 1), indicating a significantly reduced defect density on the perovskite surface. The space charge-limited current (SCLC) measurement confirms the decreased defect density from $6.60 \times 10^{15}$ to $1.07 \times 10^{15}$ cm$^{-3}$ obtained from the SBI (Supplementary Fig. 12). The enhanced PL quantum yields (PLQYs) (Fig. 3b) for the SBI-treated perovskite in comparison with the control one indicates improved optoelectronic quality of the perovskite films and suppressed trap-assisted nonradiative recombination, which is expected to enhance the $V_{oc}$ in PSCs[30]. When the SBI-modified perovskite contacts with the PCBM ETL, we find that the TRPL delivers a faster decay with the average carrier lifetime from 7.92 to 4.15 ns (Fig. 3c). We conclude that the SBI helps to boost electron extraction across interface from perovskite to PCBM due to efficient defect

passivation and improved interface energetics. The high charge transfer efficiency across the interface is critical to obtain high FF in a cell[31,32].

We now compare their ultrafast transient absorption spectroscopy (TAS). The two-dimensional (2D) pseudo-color plots of the femtosecond transient absorption (fs-TA) spectrum for the neat perovskite films (without ETL) are depicted in Supplementary Fig. 13. Both the control and SBI-modified perovskite films have characteristic ground-state bleaching (GSB) signal at about 770 nm originating from the band state filling. fs-TA spectra at selected pump-probe delay time exhibits a slower GSB recovery in the SBI-modified perovskite film, which yields a longer decay time (Supplementary Fig. 14), indicating the reduced trap-assisted recombination. For the perovskite films with ETL, similar spectra features are observed but have weaker intensity and shorter GSB recovery process due to the electron extraction (Fig. 3d, e). The SBI-modified perovskite with ETL shows a faster GSB recovery compared to the control (Supplementary Fig. 15). Tracking the respective GSB decays at 770 nm, an earlier electron extraction is achieved after SBI modification (Fig. 3f), demonstrating an enhanced electron extraction at the perovskite/ETL interface.

## Photovoltaic performance and energy loss

Motivated by the above success, we fabricate p-i-n PSCs with state-of-the-art configuration of ITO/MeO-2PACz/perovskite/SBI/PCBM/BCP/Ag (Fig. 1a), in which the self-assembled [2-(3,6-dimethoxy-9H-carbazol-9-yl)ethyl]phosphonic acid (MeO-2PACz) acts as a hole transport layer (HTL). The concentrations of MPA and PEAI are optimized to be 5

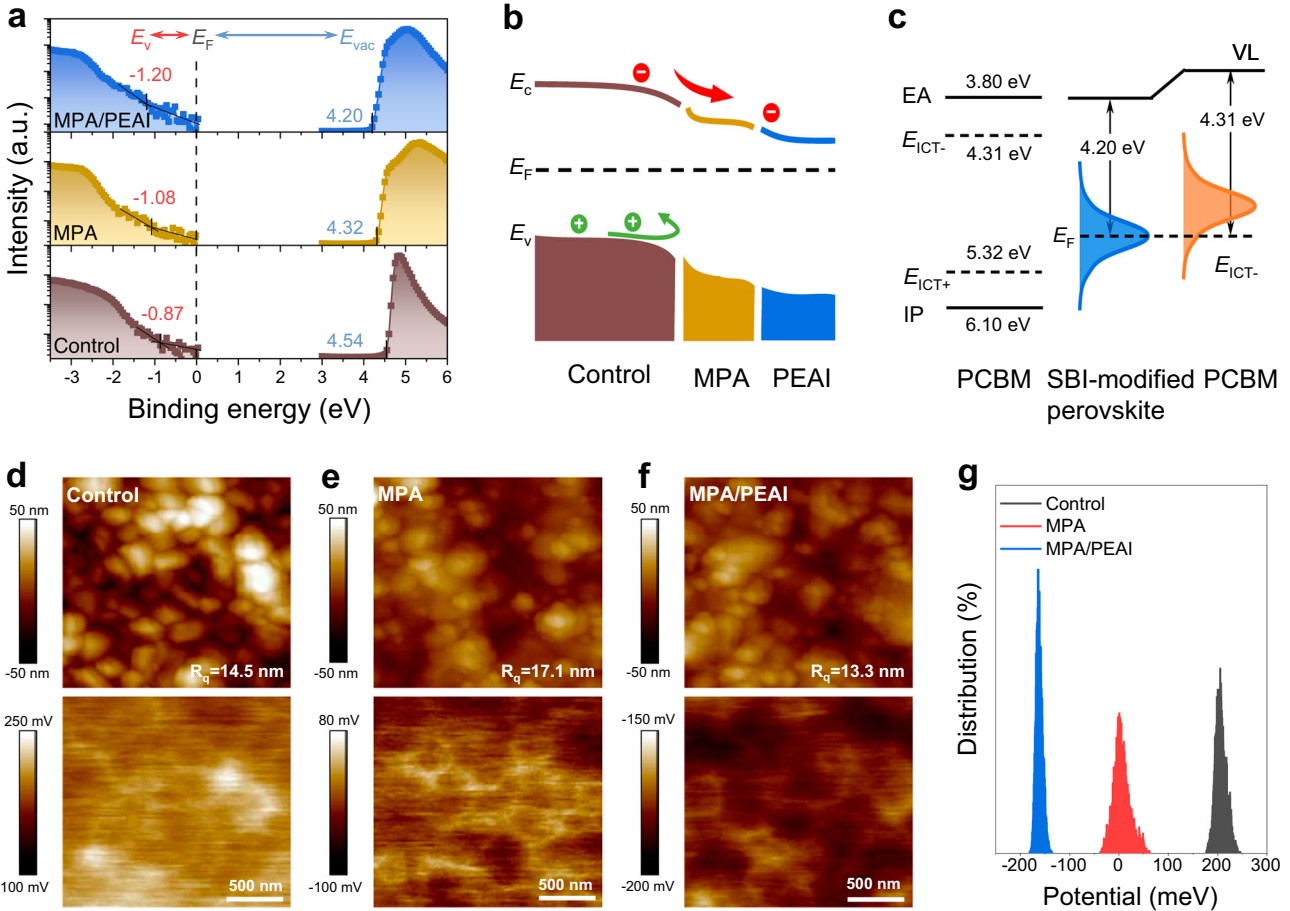

**Fig. 2 | Characterization of perovskite surface energetics. a** UPS spectra of secondary electron cutoff region and valence band region of perovskite films with and without SBI. **b** Energy level diagram of the SBI-modified perovskite. **c** $E_F$ of the SBI-modified perovskite pinning to negative polaron transport state of PCBM used as ETL. **d**–**f** Surface morphology and potential images from KPFM. **g** Statistical potential distributions of film surfaces.

and 1 mg ml$^{-1}$, respectively, to obtain the best performance (Supplementary Figs. 16, 17 and Supplementary Tables 2, 3). Figure 4a shows the current density-voltage (*J-V*) curves for control, MPA- and SBI-based PSCs under AM1.5 G simulated solar illumination and reverse scanning. The control device shows a PCE of 22.52%, with a moderate $V_{oc}$ of 1.112 V, a short-circuit current density ($J_{sc}$) of 24.91 mA cm$^{-2}$, and an FF of 81.3% (Supplementary Table 4). In contrast, the MPA-based device displays an enhanced PCE of 24.64%, with an increased $V_{oc}$ of 1.170 V, a $J_{sc}$ of 25.22 mA cm$^{-2}$, and an FF of 83.5%. Compared with the control and MPA-based devices, the PCE of the SBI-based devices further increases to 25.84% (certified 25.05%, Supplementary Fig. 18), mainly owing to improvements in $V_{oc}$ (1.194 V) and FF (84.9%). The device hysteresis is reduced for the SBI-based devices (Supplementary Fig. 19). Notably, when changing the order of MPA and PEAI, the performance of PEAI/MPA-based device shows limited improvement compared to the target SBI (MPA/PEAI)-based device (Supplementary Fig. 20), and the single PEAI-based device also has a lower efficiency than the single MPA-based device (Supplementary Fig. 21 and Supplementary Table 5), confirming the better passivation efficacy of MPA.

Corresponding external quantum efficiency (EQE) spectra in Fig. 4b yield comparable integrated $J_{sc}$ to that extracted from *J-V* curves. The SBI-based devices are measured at the maximum power point (MPP) tracking for 300 s to obtain a stabilized photocurrent of 24.31 mA cm$^{-2}$ and a stabilized PCE of 25.53% (Fig. 4c). The statistical analysis of the photovoltaic parameters based on 25 devices exhibits a satisfying reproducibility (Fig. 4d and Supplementary Fig. 22). We also examine the effect of MPA and SBI on PSCs using MA-free

$Cs_{0.05}FA_{0.95}Pb(I_{0.95}Br_{0.05})_3$, which shows increased PCEs from 21.36% (control) to 22.84% (MPA) and 23.99% (SBI), respectively (Supplementary Fig. 23 and Supplementary Table 6).

We conduct analysis of the photovoltage loss for the SBI-based devices. The improved $V_{oc}$ is concomitantly reflected in capacitance-voltage (*C-V*) plot via Mott-Schottky analysis: $\frac{1}{C^2} = \frac{2(V_{bi}-V)}{A^2\varepsilon\varepsilon_0 N_A}$, where A is device area, $\varepsilon$ is relative permittivity, $\varepsilon_0$ is vacuum permittivity, and $N_A$ is carrier concentration. The built-in potential ($V_{bi}$) increases from 1.00 (control) to 1.09 (MPA) and 1.14 V (SBI), respectively, due to the improved interface energetics (Supplementary Fig. 24). Additionally, according to detailed balance theory[33], the EQE$_{EL}$ values of 0.78%, 6.05% and 10.30% at the carrier density close to the $J_{sc}$ for the control, MPA- and SBI-based devices (Fig. 4e), indicating 126, 73 and 59 mV of nonradiative recombination induced $V_{oc}$ loss ($\Delta V_{oc,\ nonrad}$), respectively. The low $\Delta V_{oc,\ nonrad}$ of the SBI-based device is one of the smallest values reported in inverted PSCs[34,35], further demonstrating the excellent impact of SBI on reducing nonradiative recombination loss (Fig. 4f and Supplementary Table 7). The suppressed nonradiative recombination is the main factor contributing to the largely increased $V_{oc}$ of the SBI-based devices (Fig. 4g).

To gain insights on the FF loss, we perform light intensity dependent $V_{oc}$ measurement of the devices (Supplementary Fig. 27). The diode ideality factor (n), extracted from the slope of $V_{oc}$ evolution in term of $nkT/q$ are 1.73, 1.54, and 1.42 for the control, MPA-and SBI-based devices, respectively. The maximum FF (FF$_{max}$) without charge transport loss can be estimated by the equation[36]:

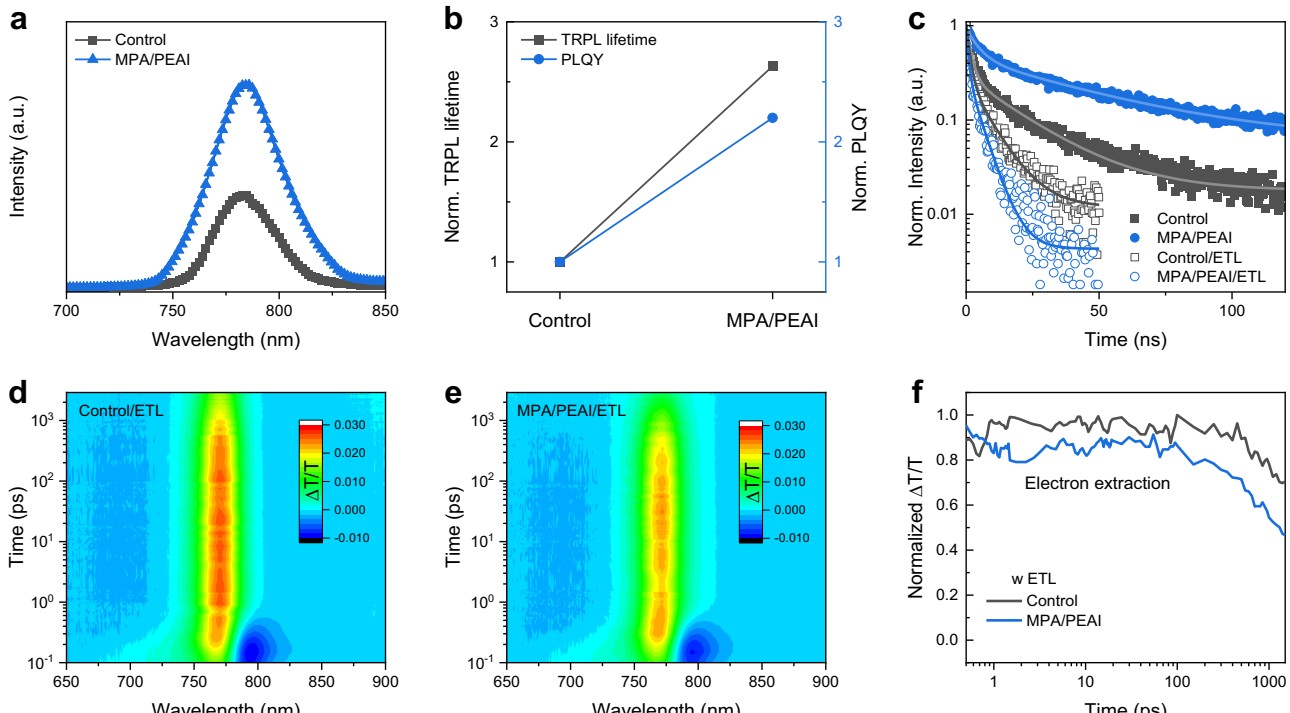

**Fig. 3 | Impact of SBI on charge carrier dynamics. a** PL spectra. **b** Normalized TRPL lifetime and PLQY of control and SBI-modified perovskite films. **c** TRPL spectra of control and SBI-modified perovskite films with ETL and without ETL. Comparison of fs-TA 2D pseudo-color plots of (**d**) control and (**e**) SBI-modified perovskite films with ETL. **f** Corresponding GSB decay at 770 nm of control and SBI-modified perovskite films with ETL.

$FF_{max} = \frac{v_{oc} - \ln(v_{oc} + 0.72)}{v_{oc} + 1}$, where $v_{oc} = \frac{qV_{oc}}{nKT}$. The calculated $FF_{max}$ of the control, MPA- and SBI-based devices are 83.53%, 85.44%, and 86.51%, respectively. The discrepancy of $FF_{max}$ is well consistent with the suppression of nonradiative recombination loss in the devices. Moreover, charge transport induced FF loss declines from 2.22% (control) to 1.92% (MPA) and 1.63% (SBI), respectively, attributed to the increased electron extraction efficiency at the perovskite/ETL interface (Fig. 4h).

We further examine the operational stability of unencapsulated devices aging $55 \pm 5\,^{\circ}C$ in dry ambient air. The SBI-based device retains about 95% of its initial efficiency after 1000 h, whereas the control device shows an over 20% efficiency loss (Fig. 4i). The SBI-modified device can maintain 88% of its initial efficiency after MPP tracking for 800 h under 1-sun illumination (Insert of Fig. 4i). We attribute the improved thermal and operational stability to the strong covalent bonding between perovskite and MPA, which not only reduces the vulnerable defect sites but also enhances the perovskite lattice structure. Moreover, the SBI-based device has no efficiency degradation after 2000 h storage in ambient air at room temperature (Supplementary Fig. 28).

## Discussion

We perform a theoretical analysis via density functional theory (DFT) calculation to better understand the mechanism of the SBI strategy on performance enhancement. We choose the main component FAPbI₃ in the calculation for simplification, and a FAI-terminated (001) perovskite surface with the predominant lattice defect of iodine vacancy ($V_I$) is constructed[37]. A noticeable interaction energy ($E_{int}$) of −0.41 eV is obtained when deprotonated MPA is placed and relaxed on the perovskite surface, revealing a thermodynamically favorable reaction with the formation of P-O-Pb bonds (2.49 Å) (Fig. 5a). For the surface without MPA treatment, the $V_I$ and exposed uncoordinated $Pb^{2+}$ induce a localized charge distribution on the perovskite surface, which generates trap states within band gap (Fig. 5b, c and Supplementary Fig. 29). In contrast, the P-O-Pb bonds induced by MPA restore the six-

coordinate local chemical environment for Pb and lead to a delocalized charge distribution with the disappearance of trap states as demonstrated in the density of states (DOS) plot, rationalizing the defect passivation ability of the in-situ surface chemical reaction. The findings from the PbI-terminated (001) perovskite surface (Supplementary Figs. 29 and 30) exhibit qualitatively consistent behaviors.

We conduct solid-state nuclear magnetic resonance (ss-NMR) spectra to consolidate the chemical reaction between perovskite and MPA. As shown in Fig. 5d, the ¹H resonance signals of FA(NH₂) and MA (CH₃) from perovskite indicate an upward chemical shift after MPA treatment, which can be attributed to the hydrogen bonding between FA⁺/MA⁺ and MPA. The introduced hydrogen bonds are expected to enhance perovskite crystal structure and beneficial for device stability. Moreover, a new chemical environment at lower chemical shift (around 28.57 ppm) in the ³¹P ss-NMR spectra is recorded form the MPA-treated perovskite (Fig. 5e), further demonstrating the reaction between perovskite and MPA in formation of Pb-O-P bonds. The chemical reaction is also expected to reconstruct the perovskite surface energetics by charge transfer and redistribution. As shown in Fig. 5f, a calculated work function difference ($\Delta WF_1$) of −1.43 eV is obtained between the control and MPA-treated perovskite surface, where $\Delta WF_1 = WF_{MPA-treated} - WF_{control}$. The reduced WF indicates a more n-type perovskite and agrees with UPS results and previous report[16]. Moreover, it is observed a further work function reduction ($\Delta WF_2$) of −0.71 eV when PEAI is deposited on the MPA-coated FAPbI₃ surface, attributed to the formation of surface dipole. Here, $\Delta WF_2 = WF_{SBI} - WF_{MPA-treated}$. These results well demonstrate the significant roles of our SBI strategy on perovskite surface properties, which show significant effectiveness on minimizing trap density and constructing beneficial perovskite surface energetics, and pave ways for the further improvement of PSCs.

In summary, we have demonstrated a SBI strategy by sequentially depositing MPA and PEAI modulators on the perovskite surface. The MPA induces a surface chemical reaction with the formation of strong

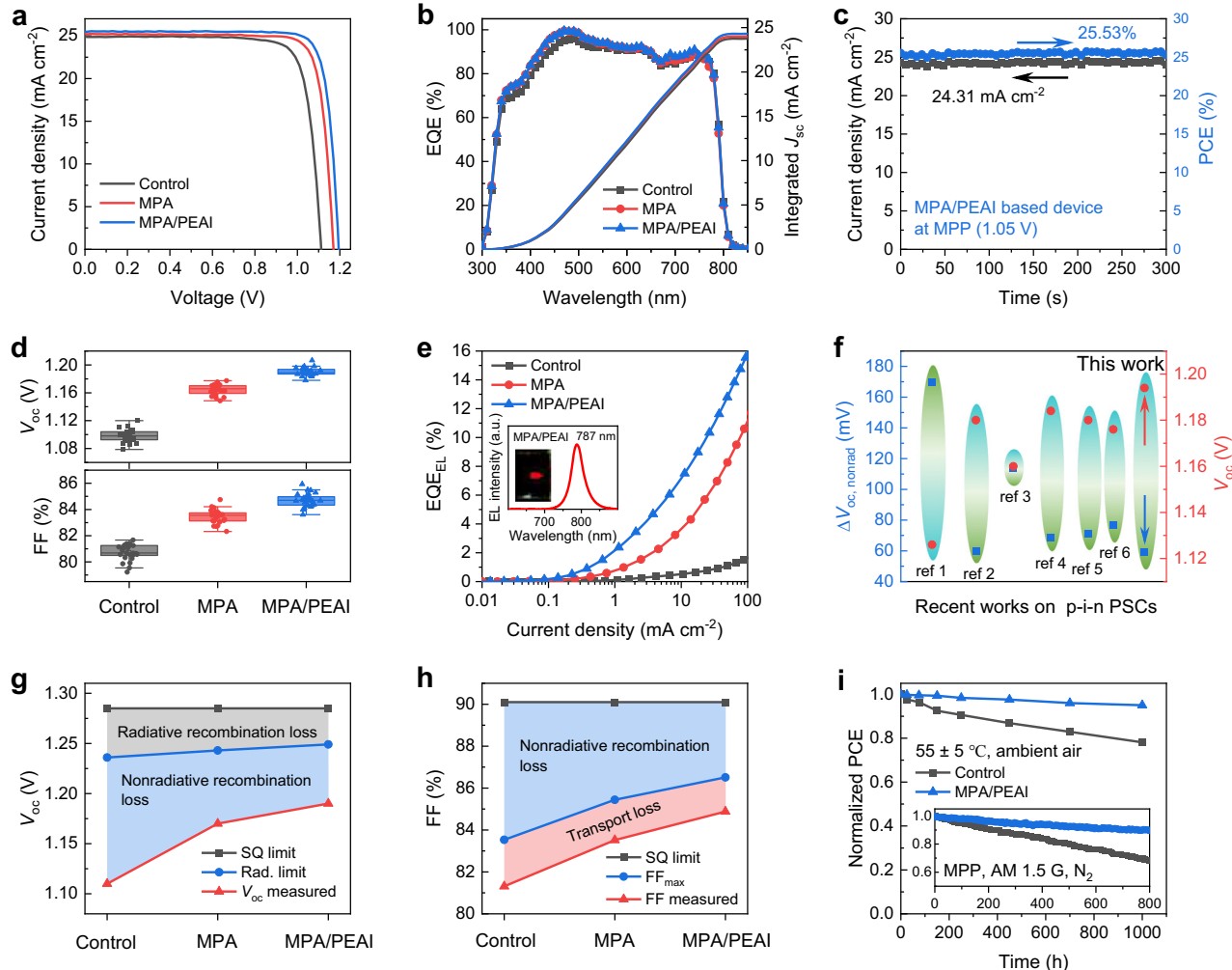

**Fig. 4 | Photovoltaic performance for PSCs. a** *J-V* curves of champion control, MPA, and SBI-based devices. **b** EQE spectra and integrated current densities. **c** Stabilized power output at the MPP for the SBI-based device. **d** Statistics of $V_{oc}$ and FF obtained from the control, MPA, and SBI-based devices. **e** EQE$_{EL}$ values of devices operating in the light emitting diode (LED) mode under different current densities. **f** $\Delta V_{oc, \text{nonrad}}$ of recent reports on p-i-n PSCs. Detailed (**g**) $V_{oc}$ loss analysis and (**h**) FF loss analysis of devices. **i** Stability of unencapsulated control and SBI-based devices aged at $55 \pm 5$ °C in ambient air. The inset represents continuous MPP tracking (in N$_2$ atmosphere).

covalent P-O-Pb bonds that provide efficient defect passivation, and the subsequent PEAI layer further modifies the surface energetics by optimizing the surface Fermi level to the PCBM ETL, leading to enhanced interfacial electron extraction. As a result, the MPA/PEAI treated p-i-n PSC achieves a stabilized efficiency of 25.53%, and one of the smallest nonradiative recombination induced $V_{oc}$ loss of only 59 mV. The certified efficiency is 25.05%. Our results suggest that cooperative interface engineering is an efficient strategy to fully modulate perovskite surface and achieve high performance p-i-n PSCs.

## Methods
### Materials
Lead iodide (PbI$_2$), formamidinium iodide (FAI), methylammonium bromide (MABr) and [6,6]-PhenylC61 butyric acid methyl ester (PCBM) were received from Advanced Election Technology Co. Ltd. Lead bromide (PbBr$_2$), methylamine hydrochloride (MACl), cesium iodide (CsI), 2-phenylethylammonium iodide (PEAI) and bathocuproine (BCP) were purchased from Xi'an Polymer Light Technology Corp. [2-(3,6-dimethoxy-9H-carbazol-9-yl)ethyl]phosphonic acid (MeO-2PACz) and 4-methoxyphenylphosphonic acid (MPA) were obtained from TCI America. Dimethylformamide (DMF), dimethyl sulfoxide (DMSO), chlorobenzene (CB), and ethanol and isopropanol (IPA) were

purchased from Sigma Aldrich. Tin oxide (SnO$_2$) colloid precursor (15% in H$_2$O colloidal dispersion) was received from J&K Scientific. All materials are used as received.

### Perovskite precursor solution and film preparation
Triple cation perovskite Cs$_{0.05}$(FA$_{0.95}$MA$_{0.05}$)$_{0.95}$Pb(I$_{0.95}$Br$_{0.05}$)$_3$ precursor (1.3 M) was prepared by mixing PbI$_2$, PbBr$_2$, FAI, MABr, and CsI in DMF/DMSO (4:1 in V/V) mixed solvent according to the stochiometric ratio. An additional 30 mol% MACl was added to the precursor. The perovskite precursor was spin-coated at 4000 rpm for 40 s with a ramp of 800 rpm s$^{-1}$, during which 200 µL CB was dropped on the film at 7 s before the end of the spin-coating. The film was then annealed at 100 °C for 30 min. MA-free perovskite Cs$_{0.05}$FA$_{0.95}$Pb(I$_{0.95}$Br$_{0.05}$)$_3$ precursor (1.3 M) was prepared by mixing PbI$_2$, PbBr$_2$, FAI, and CsI in DMF/DMSO (4:1 in V/V) mixed solvent according to the stoichiometric ratio. The perovskite film was processed with the same method as triple-cation perovskite.

### Device fabrication
Patterned indium-tin oxide (ITO) substrate was cleaned by detergent, ultra-pure water, ethanol, and isopropanol under ultrasonication for 15 min, respectively and then dried in an oven. The cleaned ITO was

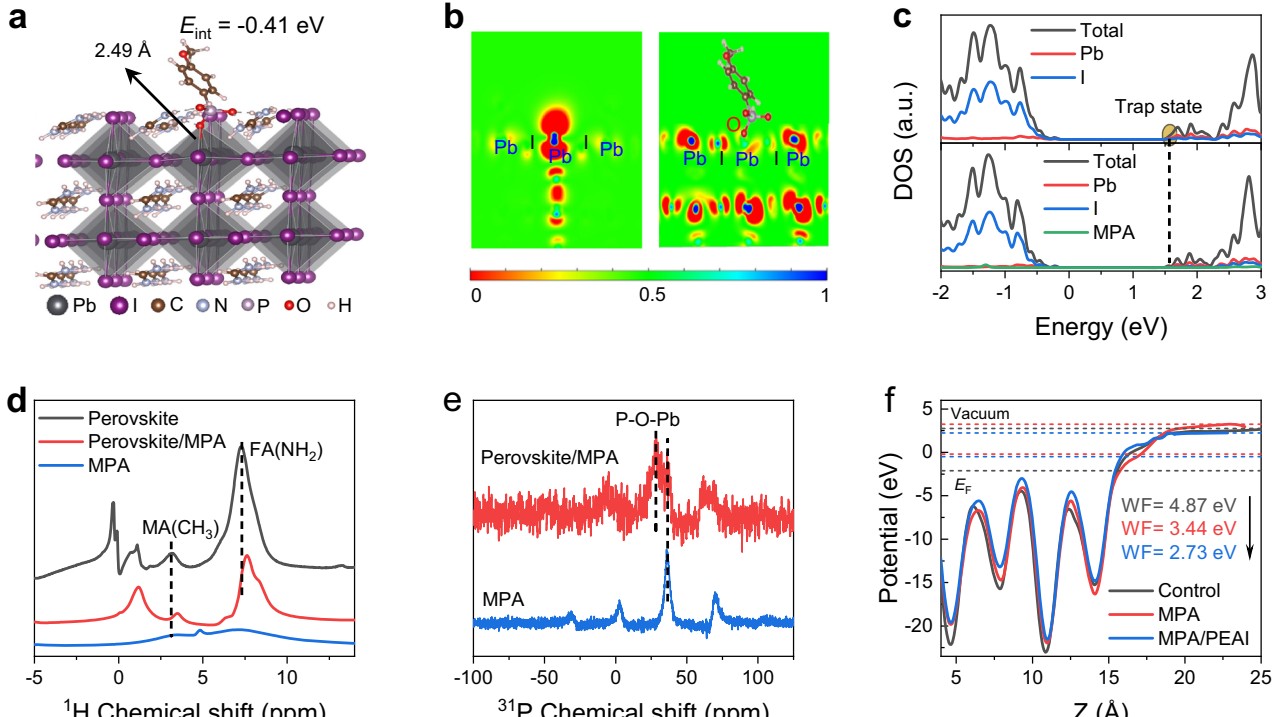

**Fig. 5 | Theoretical analysis of interaction between perovskite and SBI.**
**a** Optimized structure of MPA treated FAI-terminated perovskite (001) surface containing an iodine vacancy. **b** Calculated electron localization function and (**c**) density of states (DOS) projected onto elements of FAI-terminated surface with an iodine vacancy before and after MPA treatment. The result of the perovskite surface without iodine vacancy is shown in Supplementary Fig. 29. **d** ¹H and (**e**) ³¹P ss-NMR spectra of perovskite, MPA treated perovskite and neat MPA. **f** Work function difference (ΔWF) among control, MPA-treated, and SBI-treated perovskite surfaces.

treated by UV-ozone for 20 min before use. MeO-2PACz (0.5 mg ml⁻¹ in ethanol) as HTL was deposited on ITO at 5000 rpm for 30 s and annealed at 100 °C for 10 min. The perovskite film was coated on HTL by the abovementioned method. MPA (1–7 mg ml⁻¹ in ethanol) was deposited on the perovskite film at 4000 rpm for 30 s and annealed at 100 °C for 10 min. Then, PEAI (0.5–5 mg ml⁻¹ in IPA) was spin-coated at 4000 rpm for 30 s without annealing. Notably, MPA is not dissolved in IPA, and hence is not ruined during PEAI deposition. After that, PCBM (20 mg ml⁻¹ in CB) as ETL was spin-coated at 1800 rpm for 40 s. BCP (0.5 mg ml⁻¹ in IPA) was dynamically deposited on PCBM at 6000 rpm for 30 s. Finally, 100 nm Ag was thermally evaporated under high vacuum (<2 × 10⁻⁴ Pa). The device active area was defined as 0.05 cm².

## Film characterization
Scanning electron microscope (SEM, Zeiss G450) was performed to characterize the surface and cross-section morphology of perovskite films. KPFM was attained by Dimension Icon SPM (Bruker) with a Pt/Ir-coated tip in lift-mode. X-ray diffraction (XRD) pattern was recorded to examine perovskite crystallinity using PANalytical X-ray diffractometer with Cu Kα radiation. UV-vis absorption spectra were measured by UV-vis-NIR spectrometer (TU-1901). Steady state photoluminescence (PL) and time-resolved photoluminescence (TRPL) were obtained by a time-corrected single photon counting (TCSPC) system with an excitation wavelength of 450 nm. FTIR measurements were conducted by the PerkinElmer FTIR spectrometer equipped with an attenuated total reflectance accessory. TAS was performed with a Yb:KGW laser (1030 nm, 220 fs Gaussian fit, 100 kHz, Light Conversion Ltd) and a spectrometer (Acton SpectraPro 275) equipped with a line array CCD. ss-NMR spectra were recorded in a Bruker Avance (600 MHz) NMR spectrometer. The perovskite powders were scrapped from glass substrates which were deposited by the same method in device process, and then packed in a magic angle spinning

(MAS) rotor in N₂-filled glovebox before being transferred into the NMR probe. Photoluminescence quantum yields (PLQYs) were measured using an integrating sphere with a 450 nm excitation source and 1 sun equivalent intensity. The output of the integrating sphere was collected with a fiber connected with an Andor Kymera 193i spectrometer using a silicon CCD camera (iDus DU240A-OE). The system was calibrated with a halogen calibration lamp (HL-3 plus CAL from Ocean Optics).

## Device characterization
Current-voltage (*J-V*) curves of PSCs were recorded via Keithley 2400 system and solar simulator (SS-F5-3A, Enlitech) in the N₂-filled glovebox without any preconditioning. The light intensity (AM 1.5 G, 100 mW cm⁻²) was calibrated with a NREL certified Si cells (KG-2) to keep the spectral mismatch correction at 1.00 ± 0.01. The measurement was proceeded with a scan rate of 0.1 V s⁻¹, a voltage step of 0.02 V and delay time of 10 ms. The EQE spectra from 300 to 850 nm were collected by a QE-R system (Enlitech). The operational stability of the unencapsulated devices at the MPP tracking was tested under a commercial xenon lamp chamber with one-sun (AM 1.5G) illumination in nitrogen atmosphere. Capacitance-voltage (*C-V*) curve was conducted by an impedance spectroscope (PGSTAT302N, Autolab) with a frequency of 5 kHz. A digital oscilloscope (DOS-X 3104 A) was used to measure the transient photovoltage decay at open-circuit condition. SCLC measurement was applied to determine the electron trap density and mobility using the electron-only devices. The *J-V* curves are conducted under dark condition, which can be divided into three regions: the Ohmic region (*n* = 1), the trap-filling limit (TFL) region (n >3) and the SCLC region (*n* = 2). n refers to the slope of linear region. In the TFL region, the trap density ($N_t$) is determined by the onset of the trap filling limit voltage ($V_{TFL}$) using the equation: $V_{TFL} = \frac{eN_tL^2}{2\varepsilon\varepsilon_0}$, where $\varepsilon_0$ is the vacuum permittivity, and $\varepsilon_r$ is the relative permittivity of

perovskite. $L$ is the thickness of perovskite layer. In the SCLC region, the electron mobility ($\mu_e$) can be calculated according to the Motto-Gurney law: $J_d = \frac{9}{8}\varepsilon\varepsilon_0\mu\frac{V^2}{L^3}$, where $J$ is the current density, and $V$ is the base voltage.

### Photoelectron spectroscopy measurement
Ultraviolet and X-ray photoelectron spectroscopy (UPS and XPS) measurements were performed in ultrahigh vacuum surface analysis system (base pressure ≈ $10^{-10}$ mbar) equipped with Scienta-R3000 spectrometer. UPS employed a monochromatic He I light (21.22 eV) as the excitation source with an energy resolution of 50 meV. The work function and VBM were derived from the secondary electron cutoff and the frontier edge of the occupied density states to vacuum level, respectively. XPS was measured using the monochromatic Al Kα (1486.6 eV). All spectra were calibrated by referencing to Fermi level and Au $4f_{7/2}$ position of the Ar$^+$ ion sputter-cleaned Au foil, and Fermi level was referred as the zero BE.

### Ultrafast transient absorption spectroscopy
fs-TA spectroscopy was carried out by a Yb:KGW laser amplifier(Light Conversion Ltd), which provides 800 nm fundamental pulses with pulse width of 100 fs at 1 kHz. The fundamental pulses were separated to two light beams, where the pump beam (400 nm) was generated by an optical parametric amplifier (ORPHEUS-N, Light Conversion Ltd), the probe beam (white light continuum) was generated by focusing on a YAG plate. The pump and probe beam were spatially overlapped on the sample at a small angle less than 10°.The transmitted probe light from sample was collected by a linear charge-coupled device (CCD) array. The transient absorption spectra at selected pump-probe delay time were fitted by single-exponential equation: $y = A_1 \exp\left(-\frac{t}{\tau_1}\right)$.

### Electroluminescence efficiency measurements
The electroluminescence (EL) spectra were recorded by a Kymera-328I spectrograph and a Si EMCCD camera (DU970PBVF, Andor). EL efficiency (QE$_{EL}$) was performed via a digital source meter (Keithley 2400) for injecting current into perovskite solar cell and a picometer (Keithley 6482) with a Si photodiode for quantifying photons emitted from the device.

### Density functional theory calculations
DFT calculations were performed using the plane-wave pseudopotential method as implemented in the Vienna Ab initio Simulation Package. The electron-ion interactions were described by the projected augmented-wave pseudopotentials. The generalized gradient approximation formulated by Perdew, Burke, and Ernzerhof was used as the exchange correlation functional. Structural optimizations were obtained by fully relaxing all the atoms through total energy minimization with the residual forces on the atoms converged to below 0.05 eV Å$^{-1}$. The kinetic energy cutoff for the plane-wave basis was chosen to be 500 eV. To properly consider the long-range van der Waals (vdWs) interactions that play a nonignorable role in the hybrid perovskites involving organic molecules, the optB86b-vdW functional was adopted. The vacuum space was set to 15 Å along the out-of-plane direction of a FAI/PbI-terminated (001) FAPbI$_3$ perovskite surface to avoid interactions between periodic images. Considering the large simulation cell, we used only one k-point (gamma) to sample the Brillion zone. The interaction energy ($E_{int}$) is computed as the energy difference between FAI/PbI-terminated (001) FAPbI$_3$ perovskite surfaces containing a single iodine vacancy defect, with and without modification by MPA molecules.

### Reporting summary
Further information on research design is available in the Nature Portfolio Reporting Summary linked to this article.

## Data availability
The data that support the findings of this study are available in the Supplementary Information/Source Data file. Source data are provided with this paper.

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

## Acknowledgements

The work is financially supported by the National Key Research and Development Program of China (2022YFB3803300), the National Science Foundation of China grant (62322407, 22279034, 52261145698, 62125402, 62321166653), and Shanghai Science and Technology Innovation Action Plan (22ZR1418900). S.X. thanks the project funded by China Postdoctoral Science Foundation grant (BX20220089, 2022M720742). X.L., M.F., F.W., and F.G. acknowledge support by the Swedish Government Strategic Research Area in Materials Science on Functional Materials at Linköping University (Faculty Grant SFO Mat LiU no. 200900971). Calculations were performed in part at the high-performance computing center of Jilin University. We thank Tiankai Zhang (Linköping University) for discussion on PL and PLQY measurements.

## Author contributions

Q.B. and F.G. supervised the research project. S.X. fabricated the devices, conducted the characterizations, and wrote the manuscript. S.X. and S.J. contributed to the UPS and XPS measurements. F.T. performed the DFT calculations. A.C. performed KPFM measurements. Z.C. conducted TAS measurements. D.L. performed the PL and TRPL measurements. B.X. conducted SEM measurements. H.W. performed EQE$_{EL}$ measurements. H.Q. carried out ssNMR measurements. Y.Z. carried out FTIR measurements. F.W., Z.M., J.T., H.Z., Y.Y., X.L., L.Z., Z.S., M.F., J.C., F.G. and Q.B. contributed to the data analysis and revised the manuscript. All authors discussed the results and contributed to the manuscript.

## Funding

## Competing interests

The authors declare no competing interests.
