## [Peer Review File · Nature Communications]

Reducing nonradiative recombination for highly efficient inverted perovskite solar cells via a synergistic bimolecular interfaceREVIEWER COMMENTS

Reviewer #1 (Remarks to the Author):

Xiong et. al. demonstrates a synergistic bimolecular interlayer (SBI) strategy to passivate the interface between the perovskite PCBM, and obtained a high performance power conversion efficiency of over 25% for inverted PSCs. This is important as it shows the closing of the gap between the non-inverted record devices with more scalable inverted structures more likely to be manufactured in the future. The key driver being methodologies to reduce the loss mechanisms via recombination within the active devices and thereby maximising the spectrum of light harvesting.

I would suggest accepting the manuscript after addressing the following issues, which are needed to get better insight as to the fundamentals with regards to the reported efficiency improvements.

1. The author state that the interface between the active layer and the charge transport layers (CTL) requires passivation. The main focus of this work is the perovskite/PCBM interface, where MPA and PEAI were used together to passivate the perovskite/PCBM. However, the authors did not consider the Meo-2PACz/Perovskite HTL interface, which could also affect the overall performance of the device. Furthermore, a balanced charge transfer between perovskite/ETL and perovskite/HTL is essential for achieving high efficiency. Therefore, the author should perform some characterisations of the Meo-2PACz/Perovskite interface (e.g. PL, TRPL, SCLC, etc.) and investigate the influence of the MPA+PEAI passivation on this interface too.
2. The author claim that the PEAI could affect the surface roughness of the perovskite layer (Page 6). Therefore, the author should present the AFM data of the perovskite morphology with and without the PEAI treatment to validate the claim. Any spatial variations of morphology and voltage both laterally and vertically could have significant impact on the charge transport kinetics, and should be more closely examined. This is particularly relevant if the devices are to be scaled and manufactured in larger dimensions more suited for the non-inverted device configuration.

3. Please provide the bandgaps and the Jsc limits of the perovskite layer together with the PCE data.

4. PEAI is a universal passivation agent for perovskite layers, so the author could check the results by changing the order of MPA and PEAI fabrication to examine the effect of the passivation layer sequence to better identify the observed improvements in the device operations.

5. The detailed PLQE measurement and its analysis is supposed to be provided.

Reviewer #2 (Remarks to the Author):

In the paper "reducing nonradiative recombination for highly efficient inverted perovskite solar cells via a synergistic bimolecular interface" Xiong and co-workers present their bimolecular interface modification approach and demonstrate a significant enhancement to photovoltaic devices based on both a triple cation and FACs based perovskite system.

Overall, I think this is a thorough and comprehensive study. Whilst reports have been made of bimolecular interface modification strategies (e.g.

<https://www.science.org/doi/10.1126/science.adk1633>) the functionality of the molecules used in this work are significantly different to previous reports and therefore I think this work will be of interest to the Nature communications readership. I also think the thoroughness of the characterisation presented in this work makes it interesting to the wider perovskite and emerging semiconductor community.

Prior to publication I have one major comment and some minor comments which I would like addressed.

Major: I am not convinced by the KPFM data shown in Figure 2d and I think the signals presented are not true KPFM data, rather the noise resulting from incorrect measurements. True KPFM data should show some resemblance to the morphology of the surface therefore I would expect the grain definition present in SI Figure 11 to be somewhat visible here.

Some references demonstrating what I would expect this data to look like:

<https://pubs.acs.org/doi/10.1021/acsami.9b06418>,

<https://pubs.rsc.org/en/content/articlelanding/2022/nr/d1nr05045a>,

<https://pubs.acs.org/doi/10.1021/acsami.0c10641>.

This data needs to be re-measured correctly, so that the morphology of the perovskite surface is clear, and the manuscript adjusted accordingly. I would also suggest that the topography image channel measured whilst measuring the KPFM data is shown alongside the data for the reader, rather than in the SI.

Minor comments, please can the authors clarify the following in the manuscript:

1. Whether the devices measured in the stability tests of the unencapsulated devices (Figure 4i) were held in the light or the dark. If in the light, was this ambient or full spectrum sunlight? Was there any UV component?
2. For the PL and TAS measurements were the thin films on glass, quartz, or ITO glass substrates (or other)?
3. Which method was used to fit the TRPL data?
4. How were the PLQY measurements carried out, was this in an integrating sphere?

Reviewer #3 (Remarks to the Author):

The manuscript entitled 'Reducing nonradiative recombination for highly efficient inverted perovskite solar cells via a synergistic bimolecular interface' by Xiong et al. reports using the synergistic bimolecular interlayer (SBI) strategy to obtain high-performance p-i-n structure PSCs with 4-methoxyphenylphosphonic acid (MPA) and PEAI passivation. The introduction of MPA and PEAI showed their effects on defects passivation and optimization of surface Femi level, finally achieving a power conversion efficiency of 25.84% with excellent stability. While the PCE demonstrated in this work is impressive, I could not recommend acceptance

of this manuscript by Nature Communications owing to its weaknesses in novelty and science. My specific concerns are:

1. The research group previously published an article in *Adv. Mater.* 2023, 2309991, where they have already demonstrated the use of MPA for surface passivation and achieved high-performance inverted PSCs. Therefore, the authors are encouraged to provide additional explanations to clearly claim the novel contributions of their present work.
2. Again, in their previous work, MPA can induce the formation of surface dipole, whereas, PEAI could introduce surface dipole too in this work. Any molecules can form a dipole here? Why do the authors attribute the improvement to the dipole? If the improvement in device performance is due to the formation of a dipole, why do the authors not try to use just one molecule with a strong dipole order?
3. The use of both MPA and PEAI as passivation agents lack sufficient novelty.
4. The champion device's efficiency of almost 26% is very impressive and certification is highly recommended.
5. From the top-view SEM images, the control sample surface is flatter than the treated ones, which contradicts to the AFM results shown in supplementary Fig. 11. The authors also claim that the MPA surface treatment will form the inconsecutive coverage of the MPA layer, which in principle will increase the surface roughness. A follow-up question is whether surface roughness will affect the KPFM result. The authors should explain more.
6. For the charge carrier dynamics study in Figures 3 a and b, introducing ETL will typically result in a quenched PL signal and fast TRPL decay. Is this phenomenon due to increased non-radiative recombination loss or efficient charge extraction? How can these two processes be distinguished?
7. For the stability test, storage stability is meaningless. I would strongly recommend the authors carry out an operational stability test at high temperatures.
8. A minor note is that the grammar needs a thorough polish.

We thank the reviewers for their constructive comments and suggestions. We have carefully addressed all the comments and made the corresponding revision in the manuscript. In addition, we have further polished the manuscript to improve the quality of the manuscript. **All revisions are marked in red.**

REVIEWER COMMENTS

Reviewer #1 (Remarks to the Author):

Xiong et. al. demonstrates a synergistic bimolecular interlayer (SBI) strategy to passivate the interface between the perovskite PCBM, and obtained a high performance power conversion efficiency of over 25% for inverted PSCs. This is important as it shows the closing of the gap between the non-inverted record devices with more scalable inverted structures more likely to be manufactured in the future. The key driver being methodologies to reduce the loss mechanisms via recombination within the active devices and thereby maximising the spectrum of light harvesting.

I would suggest accepting the manuscript after addressing the following issues, which are needed to get better insight as to the fundamentals with regards to the reported efficiency improvements.

1. The author state that the interface between the active layer and the charge transport layers (CTL) requires passivation. The main focus of this work is the perovskite/PCBM interface, where MPA and PEAI were used together to passivate the perovskite/PCBM. However, the authors did not consider the MeO-2PACz/Perovskite HTL interface, which could also affect the overall performance of the device. Furthermore, a balanced charge transfer between perovskite/ETL and perovskite/HTL is essential for achieving high efficiency. Therefore, the author should perform some characterisations of the MeO-2PACz/Perovskite interface (e.g. PL, TRPL, SCLC, etc.) and investigate the influence of the MPA+PEAI passivation on this interface too.

Response: We thank the reviewer for the comments, and we agree that the MeO-2PACz/Perovskite HTL interface also affects the overall performance of the device. Therefore, we perform PL and TRPL measurements to investigate the effect of MPA/PEAI on MeO-2PACz/Perovskite interface (please see the following figure). It's found that the PL intensity is slightly enhanced with MPA/PEAI at the bottom interface without the MeO-2PACz layer (left figure), demonstrating the passivation effect. However, the PL quenching is suppressed and the TRPL lifetime decreases (middle and right figure), indicating the impeded charge transfer at the MeO-2PACz/MPA/PEAI/Perovskite interface. As discussed in the manuscript, it is attributed to the decreased work function by MPA/PEAI, which is suitable for the top perovskite/PCBM interface to boost electron extraction. However, MPA/PEAI with the decreased work function is not suitable for the hole-selective MeO-2PACz/Perovskite interface and thus blocks hole extraction due to the unmatched interface energetics.

2. The author claim that the PEAI could affect the surface roughness of the perovskite layer (Page 6). Therefore, the author should present the AFM data of the perovskite morphology with and without the PEAI treatment to validate the claim. Any spatial variations of morphology and voltage both laterally and vertically could have significant impact on the charge transport kinetics, and should be more closely examined. This is particularly relevant if the devices are to be scaled and manufactured in larger dimensions more suited for the non-inverted device configuration.

Response: We have added the AFM images of control, MPA-, SBI- and PEAI-modified perovskite films in Supplementary Figure 11. An increased surface roughness is observed after MPA treatment due to the inconsecutive coverage of MPA layer, while the further deposition of PEAI can significantly decrease the surface roughness by forming a more uniform surface, consistent with the SEM images. Moreover, the surface roughness R_q decreases from 13.1 to 11.0 nm after PEAI treatment.

Supplementary Figure 11

3. Please provide the bandgaps and the J_{sc} limits of the perovskite layer together with the PCE data.

Response: Perovskite with the bandgap of 1.55 eV is employed in this work, which corresponds to a J_{sc} limit of about 28 mA cm^{-2} (Chem. 2020, 6, 1254). We have added the information in the PCE data of Supplementary Table 4.

4. PEAI is a universal passivation agent for perovskite layers, so the author could check the results by changing the order of MPA and PEAI fabrication to examine the effect of the passivation layer sequence to better identify the observed improvements in the device operations.

Response: When changing the order of MPA and PEAI (please see the following figure), the device performance with PEAI/MPA shows limited improvement compared to the device with MPA/PEAI in the manuscript, which is due to less passivation efficacy of PEAI via secondary chemical bonding than MPA via primary covalent bonding in the

manuscript. In addition, the device with single PEAI also has a lower efficiency than the device with single MPA, confirming the better passivation efficacy of MPA layer. Therefore, we firstly deposit MPA for interface passivation via covalent bonding, then deposit PEAI to further improve interface energetics for promoting electron extraction.

5. The detailed PLQE measurement and its analysis is supposed to be provided.

Response: We have added the detailed PLQY measurement in the Methods Section: “Photoluminescence quantum yields (PLQYs) were measured using an integrating sphere with a 450 nm excitation source and 1 sun equivalent intensity. The output of the integrating sphere was collected with a fiber connected with an Andor Kymera 193i spectrometer using a silicon CCD camera (iDus DU240A-OE). The system was calibrated with a halogen calibration lamp (HL-3 plus CAL from Ocean Optics).”

We also have PLQY analysis in the manuscript: “The enhanced PL quantum yields (PLQYs) (Fig. 3b) for the SBI-treated perovskite in comparison with the control one indicates improved optoelectronic quality of the perovskite films and suppressed trap-assisted nonradiative recombination, which is expected to enhance the V_{oc} in PSCs.”

Reviewer #2 (Remarks to the Author):

In the paper "reducing nonradiative recombination for highly efficient inverted perovskite solar cells via a synergistic bimolecular interface" Xiong and co-workers present their bimolecular interface modification approach and demonstrate a significant enhancement to photovoltaic devices based on both a triple cation and FACs based perovskite system.

Overall, I think this is a thorough and comprehensive study. Whilst reports have been made of bimolecular interface modification strategies (e.g. <https://www.science.org/doi/10.1126/science.adk1633>) the functionality of the molecules used in this work are significantly different to previous reports and therefore I think this work will be of interest to the Nature communications readership. I also think the thoroughness of the characterisation presented in this work makes it interesting to the wider perovskite and emerging semiconductor community.

Prior to publication I have one major comment and some minor comments which I would like addressed.

Major: I am not convinced by the KPFM data shown in Figure 2d and I think the signals presented are not true KPFM data, rather the noise resulting from incorrect measurements. True KPFM data should show some resemblance to the morphology of the surface therefore I would expect the grain definition present in SI Figure 11 to be somewhat visible here. Some references demonstrating what I would expect this data to look like: <https://pubs.acs.org/doi/10.1021/acsami.9b06418>, <https://pubs.rsc.org/en/content/articlelanding/2022/nr/d1nr05045a>, <https://pubs.acs.org/doi/10.1021/acsami.0c10641>.

This data needs to be re-measured correctly, so that the morphology of the perovskite surface is clear, and the manuscript adjusted accordingly. I would also suggest that the topography image channel measured whilst measuring the KPFM data is shown alongside the data for the reader, rather than in the SI.

Response: We thank the reviewer for the constructive comments. The displayed KPFM data do not result from noise such as static electricity, as this would give a higher potential of about 1-2 V than the obtained one in KPFM data.

We have also re-measured the KPFM, and the clearer potential images along with the morphology of perovskite surface are updated in Figure 2d-2f. The SBI-modified perovskite has a diminished contact potential with a smaller surface potential distribution difference and lower surface roughness, which is beneficial for forming an efficient contact with the adjacent ETL to prevent nonradiative recombination.

Figure 2

Minor comments, please can the authors clarify the following in the manuscript:

1. Whether the devices measured in the stability tests of the unencapsulated devices (Figure 4i) were held in the light or the dark. If in the light, was this ambient or full spectrum sunlight? Was there any UV component?

Response: In Figure 4i, the stabilities of the unencapsulated devices are tested in the ambient light, which contains a small amount of UV component.

2. For the PL and TAS measurements were the thin films on glass, quartz, or ITO glass substrates (or other)?

Response: The films are coated on ITO glass for PL and TAS measurements, corresponding to the devices. The similar samples also have been used in the literature (Science 2023, 382, 284; Nature Commun. 2023, 14, 1819).

3. Which method was used to fit the TRPL data?

Response: The TRPL data is carefully fitted by a bi-exponential equation: $y = A_1 \exp\left(-\frac{t}{\tau_1}\right) + A_2 \exp\left(-\frac{t}{\tau_2}\right)$, where A_1 and A_2 are the amplitude fraction for each decay component, τ_1 and τ_2 represent the time constant of the two decay types, respectively. We have added the information in Supplementary Table 1.

4. How were the PLQY measurements carried out, was this in an integrating sphere?

Response: The PLQY measurements are carried out in an integrating sphere. We have added the information in the Methods Section.

Reviewer #3 (Remarks to the Author):

The manuscript entitled 'Reducing nonradiative recombination for highly efficient inverted perovskite solar cells via a synergistic bimolecular interface' by Xiong et al. reports using the synergistic bimolecular interlayer (SBI) strategy to obtain high-performance p-i-n structure PSCs with 4-methoxyphenylphosphonic acid (MPA) and PEAI passivation. The introduction of MPA and PEAI showed their effects on defects passivation and optimization of surface Fermi level, finally achieving a power conversion efficiency of 25.84% with excellent stability. While the PCE demonstrated in this work is impressive, I could not recommend acceptance of this manuscript by Nature Communications owing to its weaknesses in novelty and science. My specific concerns are:

1. The research group previously published an article in Adv. Mater. 2023, 2309991, where they have already demonstrated the use of MPA for surface passivation and achieved high-performance inverted PSCs. Therefore, the authors are encouraged to provide additional explanations to clearly claim the novel contributions of their present work.

Response: We thank the reviewer for the opportunity to clarify our novelty. Our previous work (Adv. Mater. 2023, 2309991) focuses on 3D/2D perovskite heterojunction, where MPA was mainly employed to overcome a large energetic mismatch between 3D perovskite layer and 2D perovskite layer, and thus construct an efficient 3D/2D heterojunction for highly efficient and stable inverted PSCs.

In this work, we develop a synergistic bimolecular interlayer (SBI) strategy via covalent bond and surface dipole to functionalize perovskite/ETL interface for single 3D inverted PSCs, and the mechanisms behind the in-situ reaction between perovskite and MPA and the shift of surface Fermi level are systematically investigated by experimental evidence and theoretical explanation. With the proposed SBI strategy, we achieve the impressively stabilized efficiency of 25.53% and one of the smallest nonradiative recombination induced V_{oc} loss of only 59 mV reported to date. This work sheds light on synergistic interface engineering that brings new insights for further improvement of PSCs.

2. *Again, in their previous work, MPA can induce the formation of surface dipole, whereas, PEAI could introduce surface dipole too in this work. Any molecules can form a dipole here? Why do the authors attribute the improvement to the dipole? If the improvement in device performance is due to the formation of a dipole, why do the authors not try to use just one molecule with a strong dipole order?*

Response: We fully agree with the reviewer that it is best to use just one molecule with a strong passivation and dipole effects; however, it is still a challenge to synthesize a single molecule that can provide both the desired covalent bonding and energy level alignment as discussed in the introduction section.

Surface dipole generally arises from the differences in the distribution of electron density of interfacial molecule and thus tunes surface work function, which is determined by molecular structure and arrangement.

In this work, the improvement in device performance is attributed to two issues. One is that the MPA provides efficient defect passivation with the formation of strong covalent P-O-Pb bonds and creates a more n-type perovskite surface. Another is that the subsequent PEAI layer further improves the surface energetics by creating an additional negative dipole to pin to the negative polaronic transport state of PCBM ETL, and promoting electron extraction at perovskite/PCBM interface.

3. *The use of both MPA and PEAI as passivation agents lack sufficient novelty.*

Response: As discussed in the manuscript, MPA induces an *in-situ* chemical reaction at perovskite surface via forming strong P-O-Pb covalent bonds that passivate surface defects and upshift the surface Fermi level. The primary covalent bonds have greater robustness than the secondary chemical bonds (e.g. coordination bonds, ionic bonds and hydrogen bonds) that are normally used for perovskite passivation. Although PEAI is also a good passivation agent for perovskite, here the main functionality of PEAI is to further tune perovskite surface energetics by creating negative dipole that can promote electron extraction at the contact. Therefore, we have developed a synergistic bimolecular interlayer (SBI) strategy via MPA and PEAI to functionalize the perovskite surface for reducing nonradiative recombination.

4. *The champion device's efficiency of almost 26% is very impressive and certification is highly recommended.*

Response: We have further certified the reported performance of SBI-based PSCs by Shanghai Institute of Microsystem and Information Technology (SIMIT), where a certified efficiency of 25.05% is obtained (Supplementary Figure 18).

5. *From the top-view SEM images, the control sample surface is flatter than the treated ones, which contradicts to the AFM results shown in supplementary Fig. 11. The authors also claim that the MPA surface treatment will form the inconsecutive coverage of the MPA layer, which in principle will increase the surface roughness. A follow-up question is whether surface roughness will affect the KPFM result. The authors should explain more.*

Response: The updated Figure 2d-2f and supplementary Figure 11 provide the surface morphology and potential of the perovskite films. An increased surface roughness is observed after MPA treatment due to the inconsecutive coverage, while the further deposition of PEAI can significantly decrease the surface roughness by forming a more uniform surface, consistent with the SEM images. The MPA/PEAI-modified perovskite film thus displays a smaller surface potential distribution difference and lower surface roughness.

The small surface roughness will normally not affect the surface potential during KPFM measurement with a non-contact mode, which has a spatial resolution of a few tens of nanometers (<50 nm) that is much larger than that of control, MPA- and SBI-modified perovskite films, and a potential resolution below 10 mV that is much smaller the potential difference between these samples (Small 2011, 7, 634; Adv. Funct. Mater. 2006, 16, 1407).

6. *For the charge carrier dynamics study in Figures 3 a and b, introducing ETL will typically result in a quenched PL signal and fast TRPL decay. Is this phenomenon due to increased non-radiative recombination loss or efficient charge extraction? How can these two processes be distinguished?*

Response: We agree with the reviewer that introducing ETL (PCBM) quenches PL intensity and shortens TRPL lifetime of perovskites. Importantly, in both systems with or without MAP/PEAI interlayer, the TRPL lifetime is shorter with ETL (Supplementary Table 1). Since in the system with MAP/PEAI interlayer, the ETL is introduced on top of MPA/PEAI rather than directly on perovskite, adding ETL should not introduce nonradiative recombination in perovskites. Therefore, we prefer to believe that the faster TRPL decay is due to dominant charge extraction, instead of increasing nonradiative recombination.

7. *For the stability test, storage stability is meaningless. I would strongly recommend the authors carry out an operational stability test at high temperatures.*

Response: The device stability is investigated via comparison method under the same condition, which also has been widely used in the literature (Nature Photon. 2024, 18, 379; Science 2023, 380, 823).

We fully agree that the operational stability test is better, therefore we have added the operational stability in Supplementary Figure 27.

Supplementary Figure 27

8. A minor note is that the grammar needs a thorough polish.

Response: We have polished the grammar in the manuscript.

REVIEWER COMMENTS

Reviewer #2 (Remarks to the Author):

I am satisfied with the changes to the manuscript that the authors have made due to suggestions from both myself and other reviewers, I am happy to recommend this paper for publication in Nature Communications.

Reviewer #3 (Remarks to the Author):

The authors replied to my concerns and provided strong evidence to support them. So I recommend it to be published in Nature Communications in its current state.

Some minor suggestions:

1. Include the MPP data in the main text and add the corresponding measurement details in the Methods section.
2. Update the revised version: include the certified PCE in the abstract and conclusion part.

Revision: “Reducing nonradiative recombination for highly efficient inverted perovskite solar cells via a synergistic bimolecular interface” (Research Article, NCOMMS-23-58823A)

We thank the reviewers for reviewing our revised manuscript. We have carefully addressed all the comments and made the corresponding revision in the manuscript. **All revisions are marked in red.**

REVIEWER COMMENTS

* Reviewer #2 notes the AFM images in Figure 2 and Supplementary Figure 11 are blurry and requests that they would need to be repeated to show the true morphology of the surfaces.

Response: We have re-measured the AFM and have updated the clearer images in Figure 2 and Supplementary Figure 11.

* Regarding your response to Reviewer #1's Comment 1, Reviewer #3 suggests the possibility that the top surface passivation molecules could penetrate through grain boundaries to the buried bottom interface, and recommends to check the bottom surface directly to reflect the real cases.

Response: As shown in the following figures, we respectively peel-off (Figure a) and scratch (Figure b) the perovskite films to expose their buried bottom interfaces, and then measure the components of the bottom interfaces via XPS. It is found that there is no characteristic P 2p peak of MPA on the bottom of perovskite film, which means that the top surface passivation molecules did not penetrate to the buried bottom interface.

* Regarding your response to Reviewer #1's Comment 3, Reviewer #3 suggests to provide the theoretical PCE and J_{sc} of the 1.55 eV bandgap perovskite absorber according to the S-Q limit.

Response: Perovskite with the bandgap of 1.55 eV is employed in this work, which corresponds to the theoretical PCE of 31.02% and J_{sc} of 27.26 mA cm⁻² according to

the S-Q limit, respectively (Solar Energy 2016, 130, 139). We have added the information in Supplementary Table 4.

** Regarding your response to Reviewer #1's Comment 4, Reviewer #3 recognises the importance of the deposition order of the MPA/PEAI passivation materials, and suggests to include the corresponding data in the supplementary information and provide explanations in the main text.*

Response: We agree with the comment. We have added the corresponding data as Supplementary Figure 20 and have provided explanations:

“Notably, when changing the order of MPA and PEAI, the performance of PEAI/MPA-based device shows limited improvement compared to the target SBI (MPA/PEAI)-based device (Supplementary Fig. 20), and the single PEAI-based device also has a lower efficiency than the single MPA-based device (Supplementary Fig. 21 and Supplementary Table 5), confirming the better passivation efficacy of MPA.”

Reviewer #2 (Remarks to the Author):

I am satisfied with the changes to the manuscript that the authors have made due to suggestions from both myself and other reviewers, I am happy to recommend this paper for publication in Nature Communications.

Response: We thank the reviewer for carefully reviewing our manuscript.

Reviewer #3 (Remarks to the Author):

The authors replied to my concerns and provided strong evidence to support them. So I recommend it to be published in Nature Communications in its current state.

Response: We thank the reviewer for carefully reviewing our manuscript.

Some minor suggestions:

1. Include the MPP data in the main text and add the corresponding measurement details in the Methods section.

Response: We agree with the comment. We have inserted the MPP data in Figure 4i. Additionally, we have added the measurement details in the methods section: “The operational stability of the unencapsulated devices at the MPP tracking was tested under a commercial xenon lamp chamber with one-sun (AM 1.5G) illumination in nitrogen atmosphere.”

2. Update the revised version: include the certified PCE in the abstract and conclusion part.

Response: We have added the certified PCE in the abstract and conclusion part.